



# Attention Enhanced 3D-U-Net++ Ocean Temperature and Salinity Reconstruction in the Northwestern Pacific based on Transfer Learning

Hao Wang[1,2,3], Linlin Zhang[2,4], Shuguo Yang[3], Xiaomei Yan[2,4], Zhen Li[2,4]

[1] College of Electromechanical and Engineering, Qingdao University of Science and Technology, Qingdao, 266061, China
[2] Key Laboratory of Ocean Observation and Forecasting and Laboratory of Ocean Circulation and Waves, Institute of Oceanology, Chinese Academy of Sciences, Qingdao, 266071, China
[3] College of Mathematics and Physics, Qingdao University of Science and Technology, Qingdao, 266061, China
[4] University of Chinese Academy of Sciences, Beijing, 101408, China

*Correspondence to*: Linlin Zhang (zhanglinlin@qdio.ac.cn)

**Abstract.** Real-time and accurate three-dimensional ocean temperature-salinity (T-S) field are of great significance for a deeper understanding of ocean dynamics and prediction skill improvement of numerical models. However, current ocean observations, especially those below the sea surface, still suffer from significant limitations in temporal and spatial resolution. Several neural network methods using multi-source satellite data for underwater temperature and salinity

reconstruction have been proposed, achieving real-time temperature and salinity reconstruction, but their biases relative to in-situ observations are still significant. This study focuses on the northwestern Pacific region (0–40°N, 120–160°E) and proposes an attention-enhanced three dimensional U-Net++ model, which reconstructs daily T–S fields (26 layers, 1/4° resolution, 5–2000 m depth) using real-time available sea surface temperature (SST) and sea surface height (SSH) data. The model introduces cross-scale feature aggregation and selective information gating, allowing it to emphasize temporally

coherent surface features most relevant to subsurface variability, while suppressing noise propagation and over-smoothing. By integrating 26 consecutive days of SST and SSH as inputs, the model effectively alleviates the underdetermined problem of mapping limited surface observations to full-depth structures. In addition, a two-stage transfer learning strategy is employed: the model is first pretrained using monthly SST/SSH data and the gridded Argo data to learn observation-dominated low-frequency spatiotemporal patterns, and then fine-tuned using daily SST/SSH data and the high-resolution

reanalysis to capture the meso-scale dynamic processes. Evaluation results demonstrate that the reconstructed T-S fields exhibit better agreement with in-situ T-S profiles from World Ocean Database than previous studies, both during the validation period and in long-term statistical analyses, indicating the reliability and accuracy of the proposed approach for subsurface ocean field reconstruction. The reconstructed T-S field is available at https://doi.org/10.57760/sciencedb.31950 (Wang et al., 2025).



## 1 Introduction

The three-dimensional (3D) temperature and salinity (T–S) field of the ocean, is the most fundamental parameters characterizing the marine environment. Investigations of the three-dimensional T–S field have provided crucial insights into oceanic dynamical processes, climate variability, and the evolution of marine ecosystems (Gill, 2016; Curry et al., 2003). However, the available in-situ observational data remain extremely sparse in both spatial coverage and temporal

resolution (Stewart, 2008), rendering them insufficient for resolving the fine-scale structures and evolution of meso-scale phenomena (Pauthenet et al., 2022). Consequently, the hydrological gridded data reconstructed from these observations also exhibit limited temporal and spatial resolution.

Satellite remote sensing provides global observations of ocean surface parameters (Klemas and Yan, 2014), including sea

surface height (SSH), sea surface temperature (SST), and sea surface salinity (SSS), with the advantages of high temporal and spatial resolution. However, these measurements are restricted to the ocean surface and cannot directly capture the subsurface density structure (Talley et al., 2011; Klemas and Yan, 2014). Fortunately, a strong physical coupling exists between the surface and subsurface layers of the ocean. The temperature, salinity, and current structures beneath the surface are closely linked to surface conditions through the exchange of heat and momentum within the seawater (Munk, 1950;

Huang, 2010). This intrinsic coupling makes it feasible to reconstruct subsurface T-S fields from surface observations (Fu and Davidson, 1996; Ali et al., 2004; Wu et al., 2012).

Early studies on T-S reconstruction primarily relied on statistical regression (Guinehut et al., 2012), interpolation, or physics-based models (Gilson et al., 1998; Willis et al., 2003). Although these methods have enabled the estimation of the

ocean's vertical structure to some extent, their applicability and accuracy are constrained by the discrepancy between their linear assumptions and the inherently nonlinear nature of the real ocean. For instance, reconstruction approaches based on the Surface Quasi-Geostrophic (SQG) approximation assume that surface density variations are entirely governed by temperature (Isern-Fontanet et al., 2006; Lacasce and Mahadevan, 2006), while neglecting the influence of salinity and other nonlinear factors. As a result, such methods struggle to accurately represent the complex vertical structures observed

in the actual ocean. In recent years, with the advancement of data assimilation technology, obtaining subsurface temperature and salinity fields through data assimilation has become an important approach (Chen et al., 2025; Martin et al., 2025). On one hand, satellites can provide large-scale and high-frequency sea surface information, which serves as a constraint for estimating subsurface fields; assimilating satellite observations helps improve the simulation of subsurface temperature and salinity structures (Fu, 2016). On the other hand, by combining numerical models with data assimilation methods,

existing temperature and salinity observation profiles can be directly used to further refine the simulated subsurface structures (Bellucci et al., 2007). Although satellite data assimilation has become a major means of obtaining subsurface



temperature and salinity fields and has shown good performance in surface and near-surface layers, there remain significant challenges in accurately reproducing the vertical structures of mesoscale eddies (Pilo et al., 2018; Gwyther et al., 2023).

With the rapid development of deep learning, neural networks have demonstrated great potential in the reconstruction of oceanic temperature and salinity fields due to their powerful nonlinear fitting capabilities (Xie et al., 2025; Wu et al., 2012). In recent years, various neural network–based temperature and salinity reconstruction methods have emerged, which can be broadly categorized into four types: point-to-point, surface-to-point, surface-to-surface, and surface-to-volume approaches. Point-to-point methods use sea surface data together with geographical coordinates (latitude and longitude) as

network inputs to estimate subsurface temperature and salinity at corresponding locations (Chen et al., 2022). These methods typically adopt architectures such as Long Short-Term Memory (LSTM) or Back Propagation (BP) neural networks (Smith et al., 2023; Su et al., 2022; Su et al., 2021). The surface-to-point approach uses surface data from a small surrounding region to reconstruct the subsurface temperature and salinity at the central point of that region, typically using convolutional neural networks (CNNs) (Zhao et al., 2025). For example, Meng et al. (2021) developed a 2D-CNN network

that reconstructed subsurface temperature and salinity anomalies (STA, SSA) at the regional center using 20° × 20° patches of SLA, SSTA, SSSA, and WSA. The surface-to-surface approach reconstructs subsurface temperature and salinity fields layer by layer for a given region using regional surface data (Su et al., 2019), and usually adopts a U-Net architecture (Song et al., 2024; Zhang et al., 2024). The surface-to-volume approach reconstructs three-dimensional subsurface fields (i.e., full-depth temperature and salinity) for a region directly from surface data, typically using neural networks with 3D

encoder–decoder architectures (Mao et al., 2023). Notably, this approach involves a mapping from limited input information (surface data) to a much larger output space (multi-depth fields), a process analogous to super-resolution tasks, which remains a considerable challenge.

    At present, data-driven neural network reconstruction methods are evolving toward multi-source data fusion. Early studies

primarily relied on SST and SSH, whereas later research incorporated additional inputs such as SSW and SSS, and more recently, even surface wind stress curl derived from surface wind fields. The inclusion of multiple data sources provides more effective input features, thereby improving reconstruction accuracy (Wu et al., 2012; Cheng et al., 2021; Wang et al., 2021; Yu et al., 2025; Zhao et al., 2025). However, except for SST and SSH, most other datasets exhibit a time lag of 3–7 days, posing a major challenge to the real-time reconstruction of 3D temperature and salinity fields.


    To achieve real-time large-depth reconstruction of subsurface temperature and salinity, this study proposes a method that relies solely on real-time available SST and SSH data. The method is based on an attention-enhanced 3D U-Net++ architecture, which effectively captures multi-scale spatial features and deep coupling relationships. A transfer learning

strategy is employed: the network is first pre-trained using monthly SST, SSH, and IPRC-Argo gridded temperature–
salinity products to learn real observational information and large-scale temporal patterns, and is then fine-tuned using
daily SST, SSH, and a well-recognized Global Ocean Reanalyses and Simulations Stream 2 version 4 data (GLORYS2V4)
to capture dynamic physical processes and characteristics. To address the challenge of reconstructing full-depth fields from
limited input data, a long time-series input strategy is introduced, where a sequence of past surface observations is used as
input. Results show that incorporating longer time series of surface information significantly improves the accuracy of full-
depth temperature and salinity reconstruction. The study area, spanning 0°N–40°N and 120°E–160°E, is a dynamically
active region of global significance, strongly influenced by major circulation systems such as the Kuroshio and North
Equatorial Current. The model successfully reconstructs temperature and salinity from 5 m to 2000 m. Results demonstrate
that the model achieves higher reconstruction accuracy than several widely used datasets across both validation and long-
term sequences, and shows closer agreement with WOD in-situ T-S profiles. These findings indicate that the proposed
neural network not only effectively fits the features of the training data but also captures the underlying physical laws
governing real ocean observations, which provides a feasible and efficient framework for achieving real-time three-
dimensional reconstruction of oceanic temperature and salinity fields.

The remainder of this paper is organized as follows: Section 2 presents the data and methods used in this study. The data
section introduces the neural network training data employed in this work and the comparative data used for quality analysis.
The methods section provides a detailed description of the attention-enhanced 3D U-Net++ model, the transfer learning
training strategy, and the approach of using long-term time series sea surface data as network inputs. Section 3 presents the
results and analysis, including comparative analyses of different transfer learning training strategies, the impact of using
long-term time series sea surface data as network inputs on reconstruction results, and the reliability analysis of the model-
reconstructed temperature and salinity data. Section 4 concludes the paper.

## 2 Data and Methods

### 2.1 Data

#### 2.1.1 Training Data

The SST data used in this study were obtained from the Optimum Interpolation Sea Surface Temperature (OISST) dataset
provided by the National Oceanic and Atmospheric Administration (NOAA) (Huang et al., 2021). This dataset has a global
spatial resolution of 0.25°×0.25° and a daily temporal resolution. It integrates satellite observations from multiple platforms
with various in situ measurements. The dataset is publicly available at: https://www.ncei.noaa.gov/data/sea-surface-
temperature-optimum-interpolation/v2.1/access/avhrr/.



The SSH data were provided by the Archiving, Validation, and Interpretation of Satellite Oceanographic Data (AVISO) program. This dataset is derived from the integration of multiple satellite altimeter missions, which have undergone rigorous calibration, validation, and data assimilation processes. It provides a spatial resolution of 1/8° and a daily temporal resolution. The dataset can be accessed at: https://doi.org/10.48670/moi-00148.

During the pretraining stage of the neural network, the gridded Argo product developed by the International Pacific Research Center (IPRC) was used as the target data. This dataset was generated through a variational interpolation algorithm applied to Argo float observations, with a spatial resolution of 1°×1° and a monthly temporal resolution. It is available at: https://apdrc.soest.hawaii.edu/projects/Argo/data/gridded/On_standard_levels/index-1.html.

For the fine-tuning stage, daily temperature and salinity fields from the GLORYS2V4 reanalysis dataset were employed as the ground truth. The GLORYS2V4 dataset provides global ocean reanalysis fields on a 0.25° grid, produced through the assimilation of observational data into an ocean circulation model. It enables comprehensive analysis of ocean dynamics and contributes to improved weather and climate prediction. The dataset is available at: https://doi.org/10.48670/moi-00024.

All datasets used for neural network training were uniformly interpolated in both spatial and vertical dimensions. The division of the training and validation datasets used during the training process is summarized in Table 1.

Table 1. Division of Training and Validation Datasets

| Input Data | Label Data | Resolution | Time Range | Training Set | Validation Set | Training Stage |
|---|---|---|---|---|---|---|
| SST、SSH | IPRC Argo | 1°, Monthly | 2005-2020 | 2005-2019 | 2020 | Pretraining |
| SST、SSH | GLORYS2V4 | 0.25°, Daily | 1993-2023 | 1993-2022 | 2023 | Fine-tuning |

**2.1.2 Validation data**

To objectively assess the reliability of the reconstructed products, this study employed quality-controlled T-S profile
observations from the World Ocean Database (WOD) for error analysis (Mishonov et al., 2024). The profiles provided by the XBT have been used after correction (Cheng et al., 2014). The WOD dataset is publicly available at: https://www.ncei.noaa.gov/products/world-ocean-database.



The High-Resolution Northwest Pacific Temperature Salinity Current Dataset was also utilized. This dataset integrates
historical hydrographic observations from CTD and Argo floats with recent satellite altimeter data. It is derived using the
Height Geostrophic Empirical Mode (HGEM) method (Zhang and Sun, 2012), which reconstructs three-dimensional ocean
temperature, salinity, and velocity fields based on large-scale continuous satellite altimetry observations. The dataset has a
spatial resolution of 0.25° × 0.25° and a temporal resolution of one week, and is available at:
https://msdc.qdio.ac.cn/data/metadata-special-detail?id=1456480706929647617&otherId=1456480709660139521.


The China Global Ocean Fusion Dataset 1.0 (CGOF1.0) was also introduced for comparative analysis. This dataset
extensively integrates more than 40 publicly available global ocean environmental datasets together with China's officially
released ocean data. It employs a data–physics hybrid intelligent big data methodology and a sparse observation data
migration and reconstruction technique. The dataset has a spatial resolution of 1/12° and a temporal resolution of one month.
It can be accessed at https://www.cmoc-china.cn/pages/CGOF.html.

## 2.2 Methods

### 2.2.1 3D-UNet++

The U-Net++ architecture used in this study is a deep learning network originally designed for image segmentation tasks.
It was proposed as an enhanced version of U-Net (Zhou et al., 2018), and its structure is illustrated in Fig.1 (a). U-Net++
employs an encoder–decoder architecture with deep supervision and a series of dense skip connections between the encoder
and decoder. Convolutional layers are introduced along the skip pathways to reduce the semantic gap between feature maps
of the encoder and decoder. In addition, the dense skip connections facilitate improved gradient flow throughout the
network.

The deep supervision mechanism in U-Net++ can operate in two modes: precise and fast. In the precise mode, outputs from
all sub-branches are averaged to generate the final segmentation map, while in the fast mode, only one segmentation branch
is used, which also allows for model pruning. By performing multi-level feature fusion, U-Net++ effectively utilizes
hierarchical feature information, transmitting low-level details to high-level semantic representations, thereby improving
both the accuracy and boundary sharpness of the reconstruction results.


Building on this foundation, to further enhance the model's capability in handling the complex task of temperature–salinity
reconstruction and to improve its ability to capture the importance of different local and channel-wise features, a
Convolutional Block Attention Module (CBAM) was integrated into the network. The architecture of CBAM is shown in
Fig.1 (b). CBAM aims to strengthen the representational power of convolutional neural networks by focusing on the most




relevant features across both spatial and channel dimensions (Woo et al., 2018). By combining channel attention and spatial

attention, CBAM provides a more comprehensive and effective feature extraction mechanism, thereby improving the

representational capability of the 3D U-Net++. CBAM consists of two main components: the Channel Attention Module

(CAM) and the Spatial Attention Module (SAM), whose structures are illustrated in Fig.1 (c) and (d), respectively.

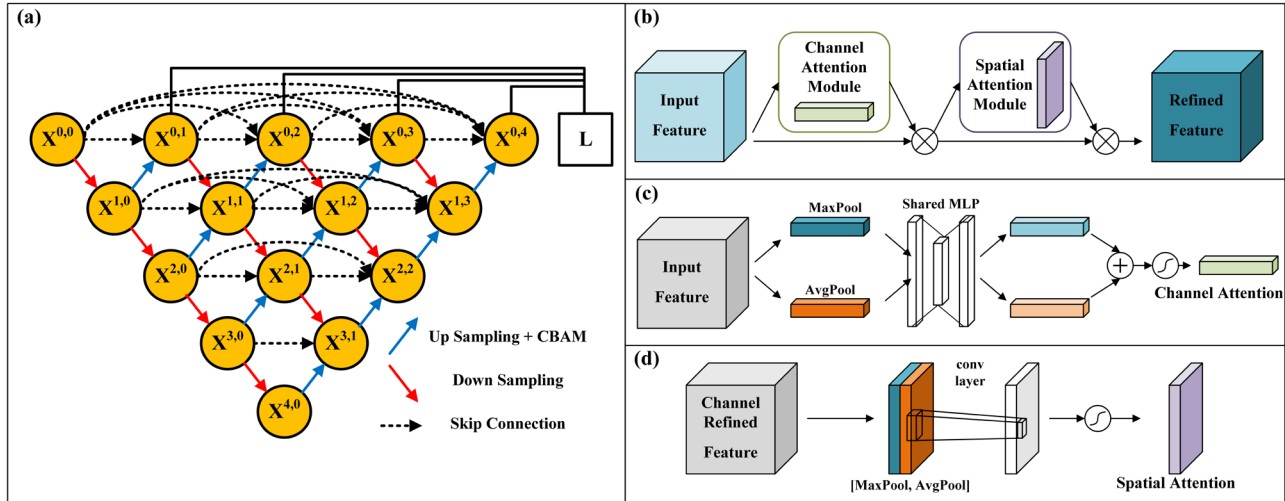

**Figure 1: Schematic diagram of the neural network structure. (a) Attention enhanced 3D-U-Net++, where $X^{i,j}$ represents the feature map at level $i$ and stage $j$. (b) Network structure diagram of CBAM. (c) CAM branch in CBAM. (d) SAM branch in CBAM.**

The CAM operates by learning which channels in the feature maps are more important, thereby enhancing the most

informative channels while suppressing irrelevant ones. This is achieved by aggregating spatial information through global

average pooling and global max pooling, followed by applying a shared multi-layer perceptron (MLP) to generate channel-

wise attention. The computational process of this module can be expressed as follows:

$$M_c(F) = \sigma\left(MLP(AvgPool(F)) + MLP(MaxPool(F))\right) \tag{1}$$

where $M_c(F)$ is output of the CAM, $\sigma()$ represents the sigmoid activation function, $MLP()$ represents the multilayer

perceptron, $AvgPool()$ and $MaxPool()$ represent mean pooling and max pooling, respectively. $F$ is the input feature.

The SAM focuses on the spatial locations of features, learning where the most important regions lie within the spatial

domain. It first applies global average pooling and max pooling along the channel dimension, concatenates the resulting

feature maps, and then performs a convolution operation to generate a spatial attention map that highlights key spatial areas.

The computational process of SAM can be expressed as follows:

$$Ms(F) = \sigma(f^{7\times7}([AvgPool(F); MaxPool(F)])) \tag{2}$$



where $M_s(F)$ is output of the CAM, $\sigma()$ represents the sigmoid activation function, $f^{7\times7}()$ denotes a convolution operation with a kernel size of 7×7, $AvgPool()$ and $MaxPool()$ represent mean pooling and max pooling, respectively. $F$ is the input feature.


Together, CAM and SAM enable the CBAM to selectively emphasize informative features, reduce irrelevant information, and enhance the discriminative capability of the model. Consequently, CBAM improves model performance across various tasks such as classification, detection, and segmentation.

### 2.2.2 Transfer Learning

To enhance the generalization capability of the model and enable it to extract features across different data modalities, thereby learning the common characteristics shared by observational and reanalysis datasets, this study employed a transfer learning strategy during neural network training.

Transfer learning is a paradigm in machine learning whose fundamental concept is to transfer knowledge acquired from
one task (the source domain) to improve learning performance on another related task (the target domain) (Pan and Yang, 2010). The transfer learning framework adopted in this study is illustrated in Fig. 2.

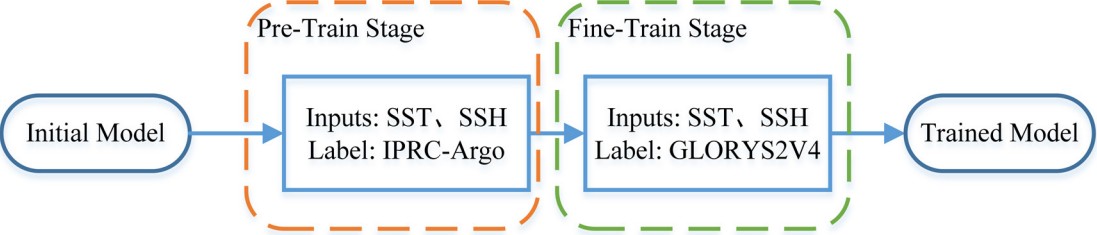

**Figure 2: Overview of Transfer Learning Strategies**

In this study, model training was conducted in two distinct stages. During the pretraining stage, monthly mean SST and SSH fields were used as input data, while the IPRC-Argo gridded temperature and salinity products served as the target labels. This stage enabled the model to learn from real observational information and capture large-scale monthly signals.
In the retraining (fine-tuning) stage, daily SST and SSH data were used as inputs, and the GLORYS reanalysis dataset was adopted as the target. This allowed the model to learn the physical patterns and data assimilation mechanisms embedded within the reanalysis data.



### 2.2.3 Long-Term Sea Surface Time-Series as Input for Model Training

This study aims to achieve high-accuracy reconstruction of subsurface temperature and salinity across 26 depth layers (5–
2000 m) using a neural network. The input data include SST and SSH, while the outputs are multi-level subsurface temperature and salinity profiles.

The main challenge of this task lies in the fact that traditional methods typically rely only on single-time (t) sea surface observations to infer the corresponding three-dimensional temperature–salinity structures, as shown in Fig. 3a. Because
the input feature dimension is relatively low while the output space is high-dimensional, the model must recover complex vertical structures from limited surface information. This imposes strict demands on the network's ability to capture and generalize spatial–temporal relationships. In essence, this problem is analogous to super-resolution reconstruction, characterized by information insufficiency and non-uniqueness.

To address this limitation, we introduce an improved strategy that incorporates temporal sequence information. Specifically,
the network input is extended from a single-time snapshot to a continuous days sequence of surface observations, spanning from t−a to t, as show in Fig. 3b. This design leverages the temporal evolution of oceanic processes, enabling the network to learn the dynamic relationships governing subsurface thermal and haline structures. In this study, considering that the number of subsurface temperature and salinity layers to be reconstructed is 26, the time-series length of the input sea surface data was also set to 26.

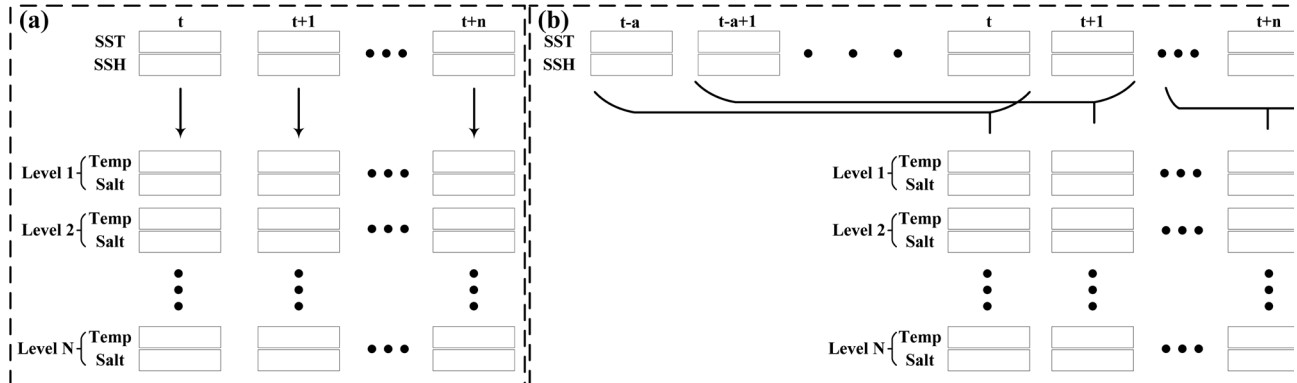

**Figure 3: Two different data input approaches. (a) Temperature and salinity reconstruction using single-day sea surface data, where $n$ denotes the total time length and $t$ represents the time step. (b) Temperature and salinity reconstruction using consecutive multi-day sea surface data, where $n$ denotes the total time length, $t$ represents the time step and $a$ represents the time window length.**



**3 Result**

**3.1 Transfer Learning**

To identify the optimal transfer learning approach, multiple transfer learning strategies were tested in this study. These included freezing only the encoder, freezing only the decoder, and freezing network weights at different percentage positions from the input to the output layers. A comprehensive analysis was performed to evaluate the effectiveness of each

strategy. The comparative results of these transfer learning strategies are presented in Fig. 4.

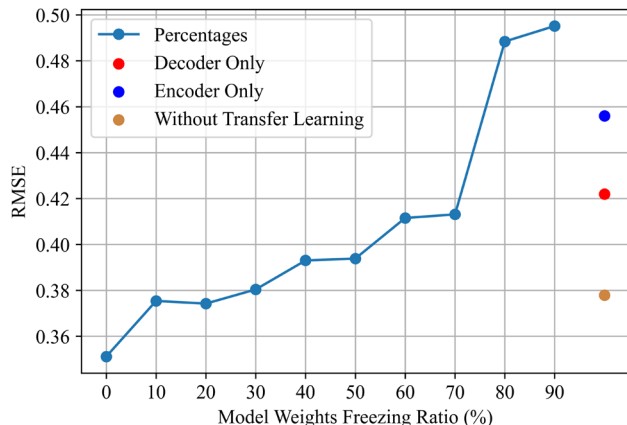

**Figure 4: Average RMSE between model outputs and WOD T-S profiles data during 2023 under different transfer learning strategies**

The root mean square error (RMSE) values shown in Fig. 4 represent the average of the temperature RMSE and salinity RMSE between model-reconstructed temperature and salinity data and WOD temperature-salinity profile observations within the study area in 2023. The results demonstrate that the reconstruction performance of the neural network varies significantly under different transfer learning strategies.


When the model was trained solely using the GLORYS2V4 reanalysis data without applying transfer learning, the baseline RMSE reached 0.3779, which is consistent with the RMSE between the 2023 GLORYS temperature–salinity data and the WOD T-S profiles (see Section 3.3.2). This indicates that when the network is trained only on GLORYS2V4 data, its overall accuracy approaches that of GLORYS itself.


When transfer learning was applied by freezing the encoder, the RMSE increased to 0.4560, whereas freezing the decoder produced an improved RMSE of 0.4219. Further analysis of gradually freezing network layers from the input to the output revealed a non-monotonic pattern. The lowest RMSE of 0.3512 was obtained when no layers were frozen, indicating full adaptability during fine-tuning. When 10 %–30 % of the network was frozen, the model achieved relatively good



performance with RMSE values close to the baseline. However, when more than half of the network parameters were frozen, the RMSE increased sharply.

These findings underscore the necessity of maintaining a balance between feature retention and model adaptability in transfer learning. The superior performance achieved without layer freezing suggests that full fine-tuning has enabled the

model to effectively capture the realistic WOD T-S profile characteristics contained in the IPRC dataset while simultaneously adapting to the physical dynamics represented by the GLORYS2V4 reanalysis data. The improved performance observed when the decoder was frozen, compared with encoder freezing, indicates that preserving the pretrained decoder's mapping capability is advantageous for reconstructing fine-scale subsurface structures. Conversely, excessive parameter freezing has been found to constrain the model's adaptability, thereby reducing reconstruction

accuracy. Overall, these results confirm that transfer learning substantially enhances cross-dataset generalization; however, its effectiveness depends on maintaining sufficient network flexibility during retraining. In this study, a transfer learning strategy without weight freezing was adopted, in which the model was fine-tuned on GLORYS2V4 data after pre-training.

### 3.2 Two Difference Input Strategies

Two input strategies were compared for subsurface temperature and salinity reconstruction: using single-time sea surface

information (single day) and incorporating a 26-day continuous sea surface sequence (26 days). The evaluation comprises two components. Firstly, the mean root mean square error (RMSE) is calculated between the model-reconstructed temperature and salinity data for 2023 and the GLORYS2V4 temperature and salinity data. Secondly, the mean RMSE is calculated between the model-reconstructed temperature and salinity data for 2023 and the WOD temperature and salinity profiles. The results are presented in Table 2.

**Table 2: Comparison of reconstruction performance under different input strategies**

| Data Input | RMSE with GLORYS2V4 | RMSE with WOD | Variable |
|---|---|---|---|
| Single Day | 0.0854 | 0.6356 | Temperature |
|  | 0.0242 | 0.0987 | Salinity |
| 26 Days | 0.0676 | 0.6096 | Temperature |
|  | 0.014 | 0.0927 | Salinity |

As shown in Table 2, the introduction of a 26-day continuous sea surface data sequence significantly improves the reconstruction accuracy of the model, indicating that temporal information can effectively enhance subsurface temperature and salinity inference. Taking GLORYS2V4 dataset as the validation data, the improvement is more pronounced: the RMSE of temperature decreases from 0.0854 °C to 0.0676 °C, and that of salinity decreases from 0.0242 PSU to 0.0140



PSU. Taking the WOD T-S profiles as the validation data, the temperature and salinity RMSEs also decrease from 0.6356 °C to 0.6096 °C and from 0.0987 PSU to 0.0927 PSU, respectively.

Overall, the improvement in salinity is generally greater than that in temperature, suggesting that salinity exhibits stronger temporal continuity and that the network can extract more stable spatiotemporal patterns from multi-day surface sequences.

Although the absolute errors for WOD T-S profiles are higher than those for GLORYS2V4, the consistent improvement across both datasets confirms the robustness and effectiveness of the neural network method based on time-series sea surface inputs.

### 3.3 Model Performance

In this subsection, the SST and SSH validation datasets in 2023 were utilized as model inputs, and the resulting model

outputs were compared with the corresponding temperature and salinity fields from the GLORYS2V4 reanalysis dataset of the same year to assess the model's performance. Furthermore, the model's reliability was validated using T-S profiles observations obtained from the WOD.

### 3.3.1 Comparison of the Model Reconstructions with GLORYS2V4

The data from 2023 were selected as an independent validation set in this study. The reconstructed temperature and salinity

fields were compared with the GLORYS2V4 reanalysis data, and the RMSE and correlation coefficient were calculated for different months and depths. Fig. 5 and Fig. 6 respectively present the vertical profiles of RMSE and correlation coefficients for temperature and salinity, with curves for different months illustrating seasonal variations and vertical characteristics.

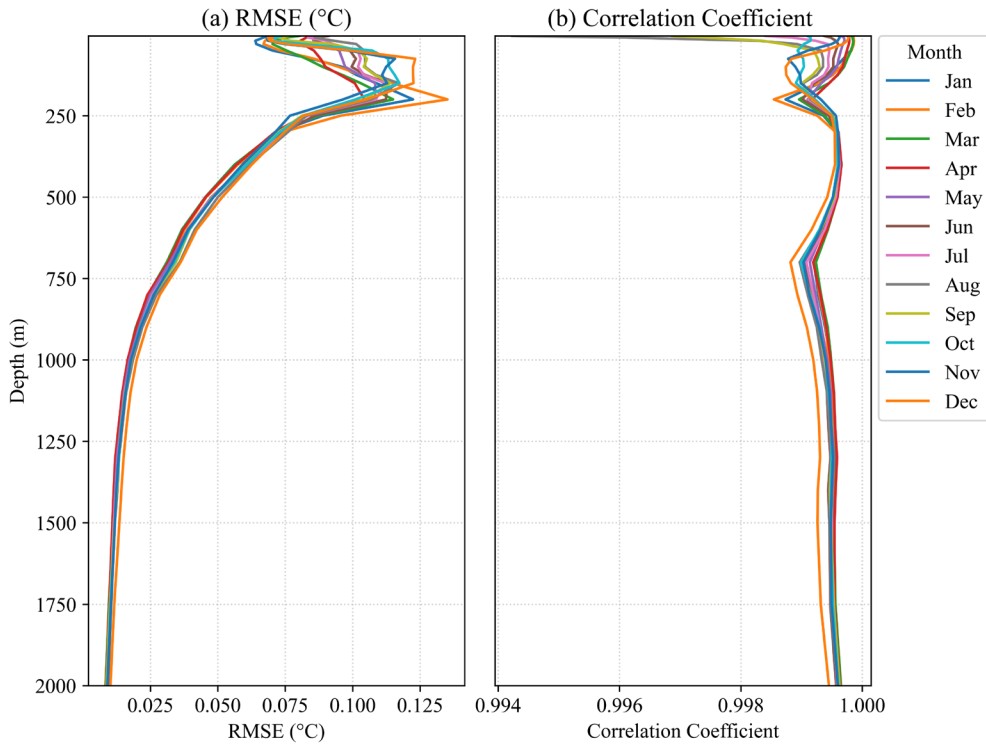

**Figure 5: (a) RMSE and (b) correlation coefficient between the model-reconstructed temperature fields and the GLORYS2V4 temperature data in 2023.**

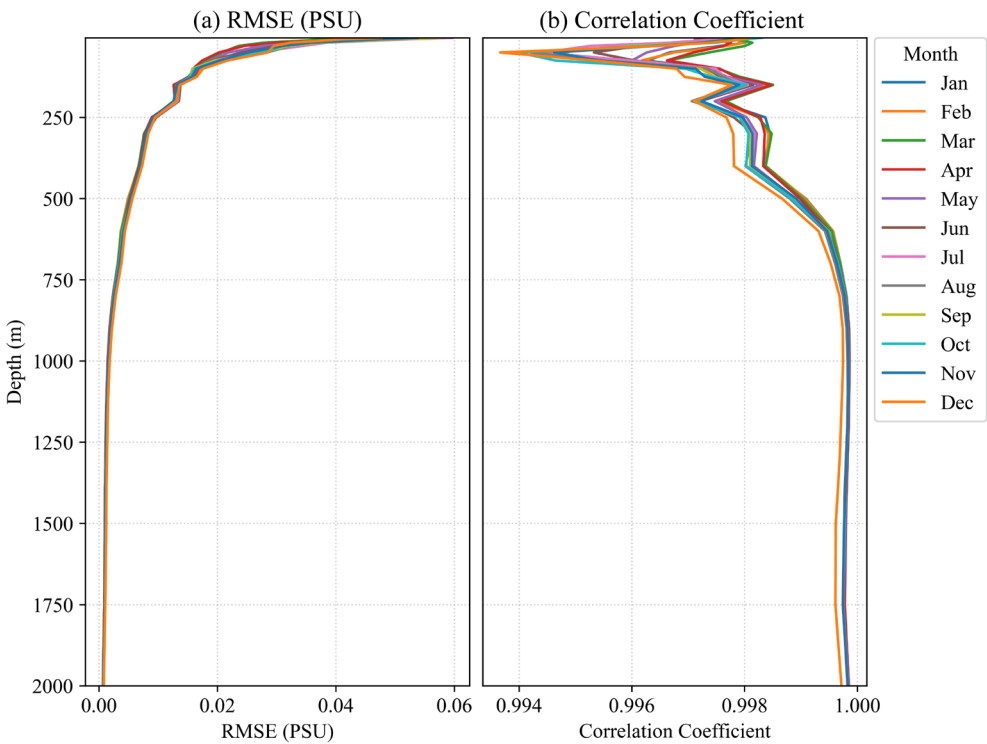

**Figure 6: (a) RMSE and (b) correlation coefficient between the model-reconstructed salinity fields and the GLORYS2V4 salinity data in 2023.**

From the RMSE distribution of temperature reconstruction (Fig. 5a), the overall RMSE ranges between 0.025°C and 0.125°C. Above 250 m, the temperature RMSE increases with depth, while below 250 m it gradually decreases, stabilizing at depths greater than 1000 m with relatively low values (approximately within 0.015°C). This trend indicates that the neural network model effectively captures the stable structure of deep-sea temperatures, though certain deviations persist in the subsurface layers influenced by atmospheric forcing and mixed-layer dynamics. The correlation coefficients (Fig. 305 5b) remain above 0.99 across all depths, demonstrating strong consistency and indicating that the model preserves the linear response relationship of temperature variations well.

The reconstructed salinity RMSE (Fig. 6a) generally decreases with depth, with surface RMSE values ranging from 0.02 to 0.06 PSU. In deeper layers (>1000 m), the RMSE markedly decreases, approaching 0 PSU, suggesting that the model 310 accurately reproduces the salinity structure in deep ocean regions. The overall correlation coefficients remain above 0.99 (Fig. 6b), showing slight fluctuations in the upper layers but minimal variation among months, reflecting the model's strong capability to reproduce the spatiotemporal distribution patterns of salinity.

**Figure 7: Reconstructed temperature (left) and GLORYS2V4 temperature (middle) and their differences (right) at depths of 50 m, 200 m, 800 m, and 1000 m on January 1, 2023.**





**Figure 8: Reconstructed salinity (left) and GLORYS2V4 salinity (middle) and their differences (right) at depths of 50 m, 200 m, 800 m, and 1000 m on January 1, 2023.**

Figure 7 and Figure 8 respectively present the comparison results between the reconstructed temperature and salinity fields

for 2023 and the GLORYS2V4 reanalysis data at four representative depths: 50 m, 200 m, 800 m, and 1000 m. Overall, the reconstructed temperature and salinity fields exhibit spatial patterns that are highly consistent with those of the GLORYS2V4 data, successfully reproducing the latitudinal gradients and large-scale thermohaline structures.

The error maps reveal that the magnitude of the discrepancies is limited and relatively uniform across the study region,

with no areas of concentrated error and near-zero differences in most regions. At greater depths, such as 800 m and 1000 m, the errors further decrease, and no systematic bias is observed, indicating the enhanced stability and reliability of the model in deep-ocean environments.

These findings confirm that the proposed neural network can effectively infer subsurface thermal structures using only

surface observational data. The high spatial consistency and low reconstruction errors across all depths highlight the model's strong generalization capacity and physical coherence. The particularly robust performance in the intermediate and deep layers underscores its ability to capture stable, large-scale oceanic states.

### 3.3.2 Comparison of the Model Reconstructions with WOD T-S Profiles

To objectively evaluate the reliability of the reconstructed products, WOD T-S profiles data were introduced for error

analysis. After quality control, a total of 7,833 T-S profiles were selected within the study region for the year 2023, as shown in Fig. 9a.

At the geographical coordinates corresponding to each T-S profile, reconstructed temperature and salinity data as well as GLORYS2V4 temperature and salinity data were extracted to compute the RMSE. The vertical RMSE distributions between the WOD T-S profiles and the reconstructed and GLORYS2V4 temperature data are shown in Fig. 9b, while the

vertical RMSE distributions between the salinity profiles and the reconstructed and GLORYS2V4 salinity data are presented in Fig. 9c.

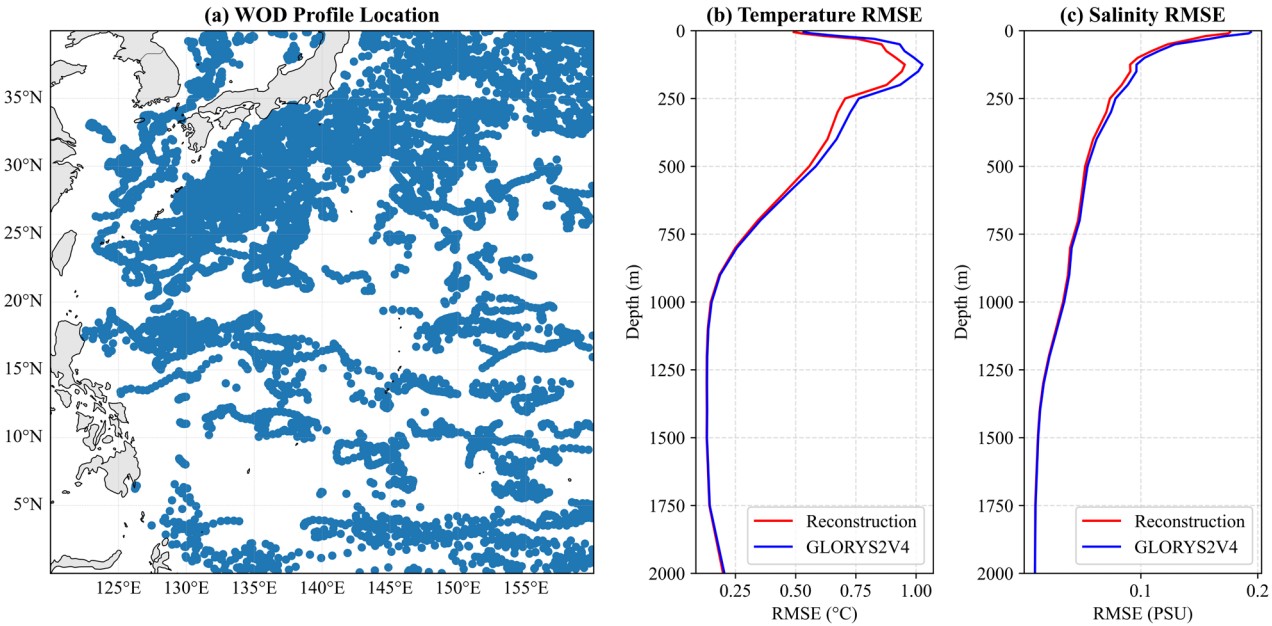

**Figure 9: (a) Spatial distribution of WOD observational profiles in 2023; (b) variation of temperature RMSE with depth between the reconstructed temperature data and the WOD temperature profiles; (c) variation of salinity RMSE with depth between the reconstructed salinity data and the WOD salinity profiles.**

In 2023, the T-S profiles distribution within the study region was relatively uniform (Fig. 9a). Overall, the reconstructed temperature and salinity data and the GLORYS2V4 data exhibit similar vertical RMSE distributions (Figs. 9b and 9c). For temperature, both datasets show higher RMSE values in the subsurface layers (Fig. 9b), while for salinity, the RMSE values

decrease gradually with depth (Fig. 9c). The reconstructed temperature and salinity data generally outperform the GLORYS results at all depths. The mean RMSE of the reconstructed temperature over the full depth range is 0.4907 °C, lower than that of GLORYS (0.5241 °C). Similarly, the mean RMSE of the reconstructed salinity (0.0699 PSU) is slightly lower than that of GLORYS (0.0747 PSU). Overall, the reconstructed temperature and salinity fields show closer agreement with the in situ observations, indicating better reconstruction fidelity.

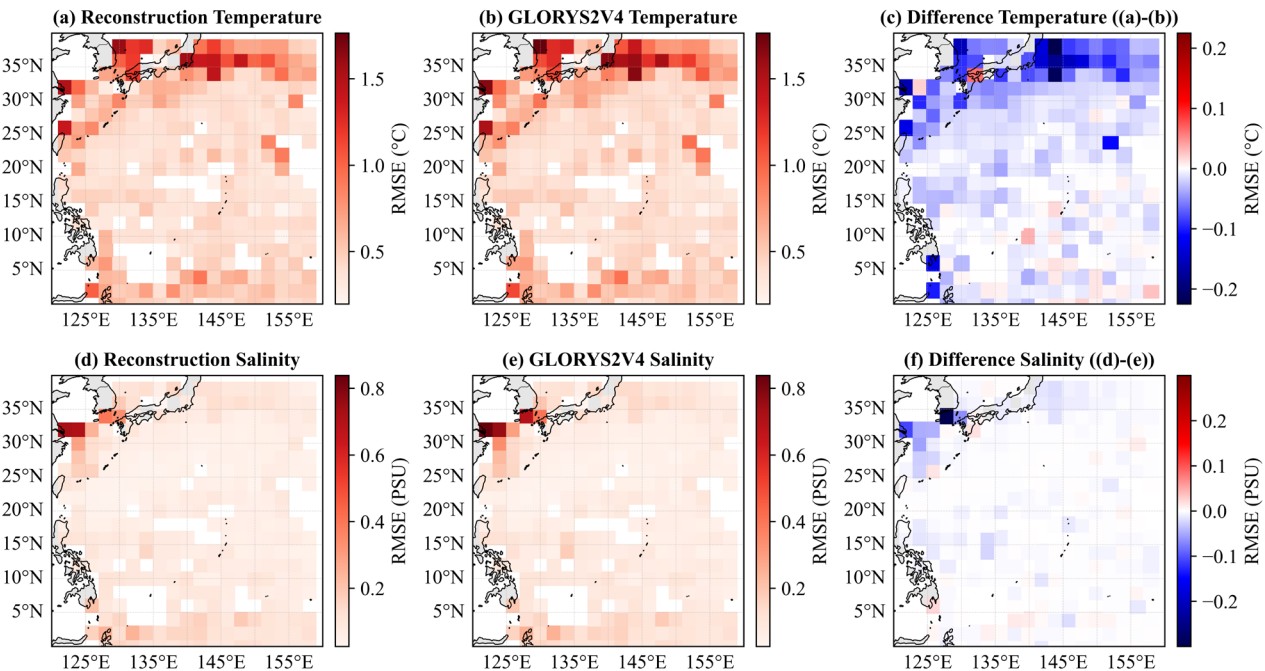

**Figure 10: RMSE between reconstruction and WOD profiles (left), RMSE between GLORYS2V4 and WOD profiles (middle), and their difference (right). The first row shows the results for temperature data, and the second row shows the results for salinity data.**

The spatial distribution of the RMSE between the reconstructed data and the WOD profiles is shown in Fig. 10. Spatially (Fig. 10a, b, d, and e), both datasets exhibit larger errors along the western boundary currents and near-coastal regions, while the errors are relatively smaller in the interior of the basin. The temperature difference map (Fig. 10c) indicates that the reconstructed temperature generally performs better across the study region, whereas the salinity difference map (Fig. 10f) shows that the reconstructed salinity slightly outperforms the GLORYS data. Overall, the reconstructed temperature

and salinity fields derived from the neural network are more consistent with the WOD T-S profiles, demonstrating higher reliability.

All the above results indicate that, by incorporating transfer learning as a training strategy, the proposed neural network–based temperature and salinity reconstruction method not only achieves a good fit to the GLORYS reanalysis data but also produces reconstructions that are even closer to the in situ observations.


To evaluate the relationship between the reconstructed temperature–salinity fields and the WOD temperature–salinity profiles, and to compare them with GLORYS2V4, correlation density scatter plots of temperature and salinity were

generated at four representative depths (50 m, 200 m, 800 m, and 1000 m). The correlation density scatter plots for temperature are shown in Fig. 11, and those for salinity are presented in Fig. 12.

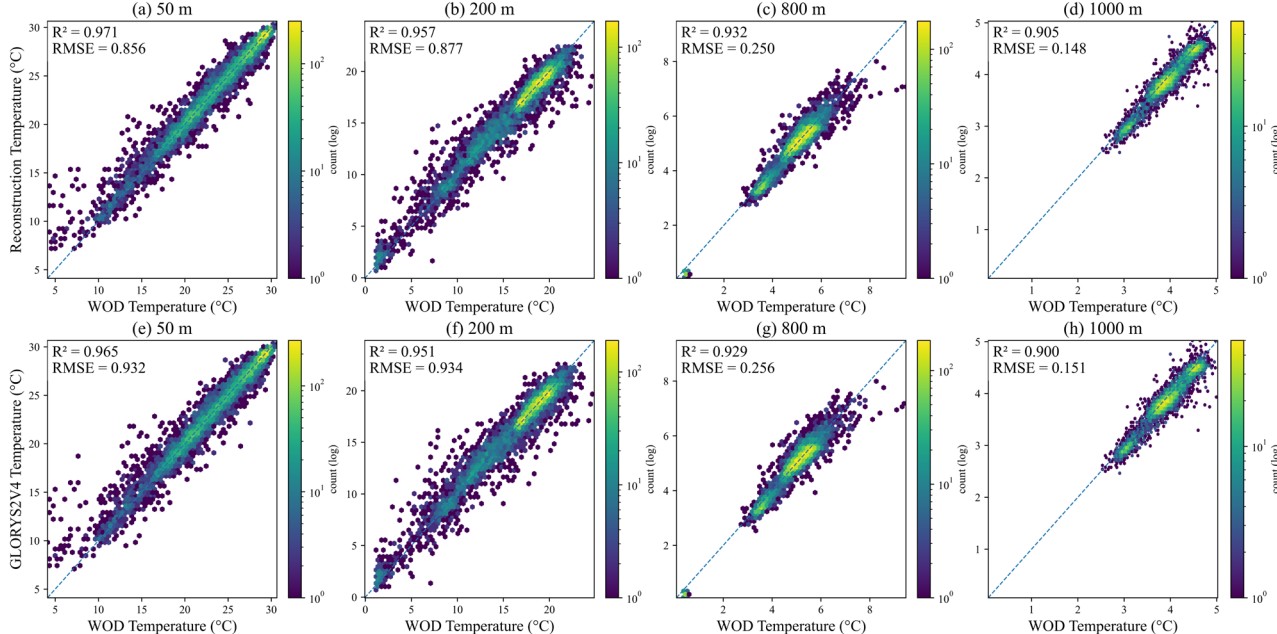

**Figure 11: Density scatter plots of reconstructed temperature versus WOD observations at different depths in 2023: (a) 50 m, (b) 200 m, (c) 800 m, and (d) 1000 m; and density scatter plots of GLORYS2V4 temperature versus WOD observations at different depths in 2023: (e) 50 m, (f) 200 m, (g) 800 m, and (h) 1000 m.**


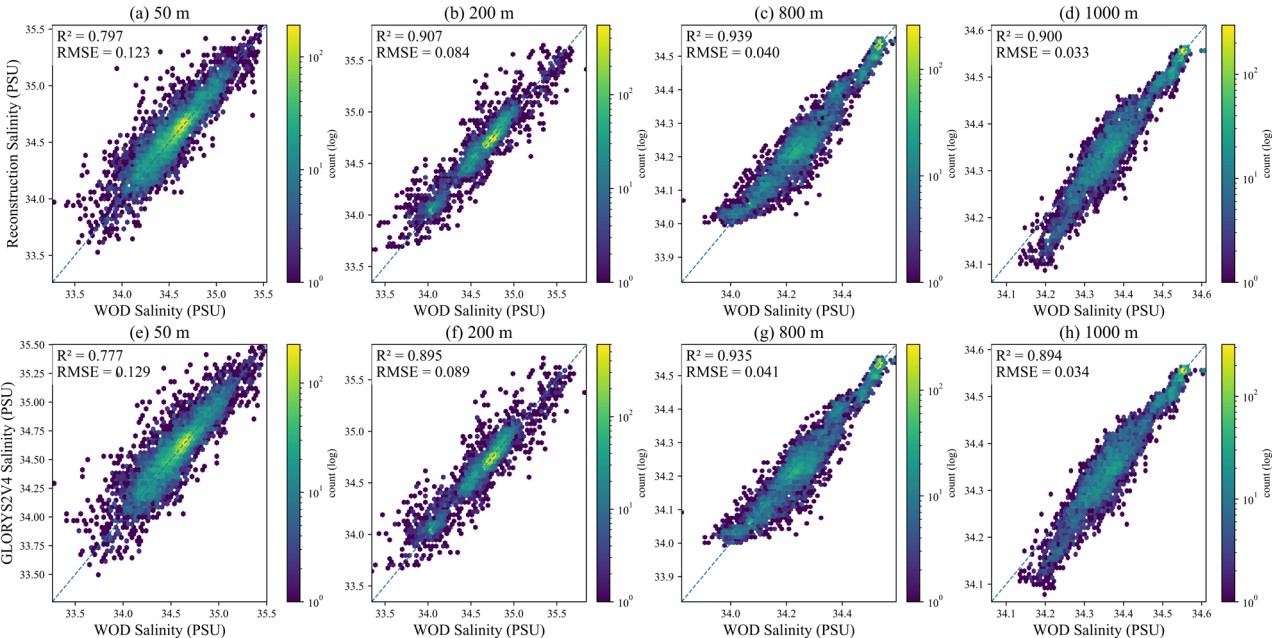

**Figure 12: Density scatter plots of reconstructed salinity versus WOD observations at different depths in 2023: (a) 50 m, (b) 200 m, (c) 800 m, and (d) 1000 m; and density scatter plots of GLORYS2V4 salinity versus WOD observations at different depths in 2023: (e) 50 m, (f) 200 m, (g) 800 m, and (h) 1000 m.**

The temperature density scatter plots (Fig. 11) and salinity density scatter plots (Fig. 12) illustrate the correspondence between the reconstructed temperature and salinity results and the WOD in situ observations at different depths (50 m, 200 m, 800 m, and 1000 m), in comparison with the GLORYS2V4 product. As shown in Fig. 11 and Fig. 12, both temperature and salinity reconstructions exhibit strong linear correlations with the WOD observations, with most scatter points densely
distributed along the 1:1 diagonal line. This indicates that the model successfully reproduces the thermohaline characteristics of oceanic water masses at all depth levels.

In contrast, GLORYS2V4 shows larger deviations in the upper layer, particularly at 50 m, with RMSE values significantly higher than those of the reconstructed results. The reconstruction model maintains higher $R^2$ and lower RMSE values in
both the surface and intermediate layers, suggesting a better fit to the observed data. As depth increases, the correlation of the reconstructed temperature and salinity further improves and the errors decrease, reflecting greater reconstruction stability in the deeper ocean. Overall, the proposed reconstruction method achieves higher accuracy than GLORYS2V4 at all depths and captures the observed thermohaline variations more effectively.


### 3.4 Long-term Statistical Analysis

To further verify the robustness and reliability of the proposed reconstruction method in generating temperature and salinity data, a long-term statistical analysis of the reconstructed results was conducted. Specifically, after the neural network training was completed, SST and SSH data from 1993 to 2023 were used as inputs to reconstruct the corresponding temperature and salinity fields for the same period. The reconstructed results were then compared with all available WOD T-S profile observations from 1993 to 2023 to perform an error analysis. For the long-term comparison, four reference

datasets—GLORYS2V4, HGEM, IPRC Argo, and CGOF—were employed.

During the period from 1993 to 2023, a total of 353,154 WOD T-S profiles were collected within the study region. For clarity of presentation, the RMSE values calculated from these 353,154 samples were averaged on a monthly basis. The RMSE results for temperature are shown in Fig. 13, and those for salinity are presented in Fig. 14. Since the valid time

span of the IPRC Argo dataset is 2005–2020, the results were divided into three distinct periods to better demonstrate the effectiveness of the transfer learning strategy: 1993–2004 (Fig. 13a and Fig. 14a), 2005–2019 (Fig. 13b and Fig. 14b), and 2020–2023 (Fig. 13c and Fig. 14c). The variation of RMSE with depth was also calculated based on the 353,154 samples, and the results are shown in Fig. 15.

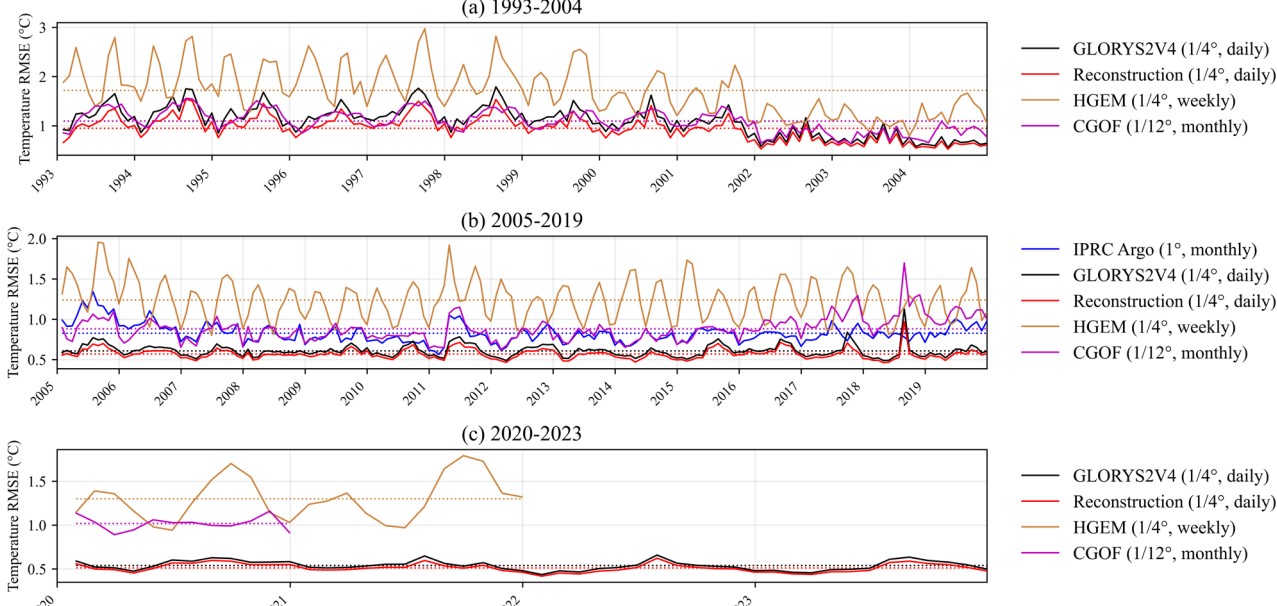

**Figure 13: Monthly time series of RMSE of various temperature datasets compared with WOD temperature profiles in the upper 2000m: (a) 1993–2004, (b) 2005–2019, and (c) 2020–2023.**



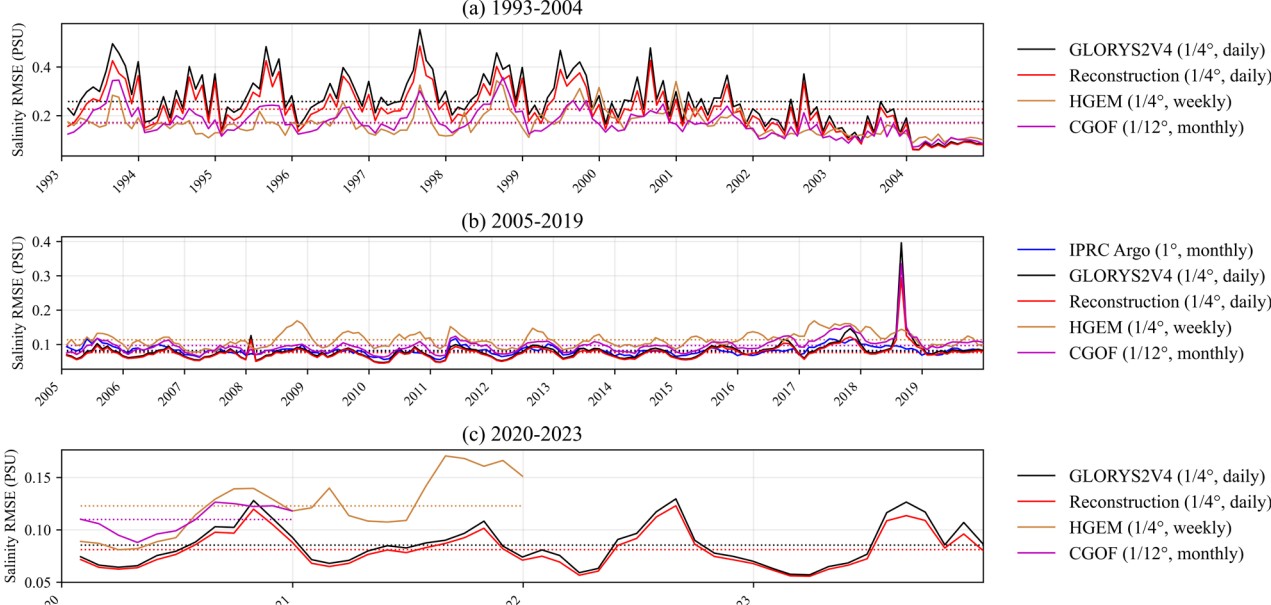

**Figure 14: Monthly time series of RMSE of various salinity datasets compared with WOD temperature profiles in the upper 2000m: (a) 1993–2004, (b) 2005–2019, and (c) 2020–2023.**

As shown in Fig. 13, throughout the entire study period (1993–2023), the reconstructed temperature field remains generally consistent with GLORYS2V4. Compared with all other reference datasets, the reconstructed temperature results are closer to the WOD observed temperature profiles (Average RMSE: 0.677267 °C), demonstrating higher accuracy and reliability.

   The RMSE results for salinity are presented in Fig. 13. During 1993–2004, HGEM exhibits the smallest errors (0.1681PSU)
among the compared datasets (Fig. 13a); however, in the subsequent periods, the reconstructed salinity data align more closely with the WOD salinity profiles, indicating better stability and adaptability of the proposed method in salinity reconstruction. Throughout the entire period (1993-2023), the reconstructed salinity shows the closest agreement with the WOD observed salinity, with an RMSE of 0.1282 PSU.

It can also be observed that the RMSE values of all datasets are relatively high during 1993–2004. This is mainly because the number of WOD temperature–salinity profiles during this period was limited, and most observations were concentrated along the Japanese coast. This region lies within the highly dynamic Kuroshio Current zone, where the vertical thermohaline structure exhibits strong cross-front differences and prominent mesoscale to submesoscale variability, resulting in larger assimilation errors between model outputs and observations.




Notably, in the proposed transfer learning strategy, the pretraining data were derived from the 2005–2019 IPRC Argo temperature–salinity dataset. Nevertheless, the reconstruction results show superior reliability compared with GLORYS2V4 over the entire 1993–2023 period. This demonstrates that the model's learning capability is not restricted to the temporal range of the training data and possesses strong generalization ability.

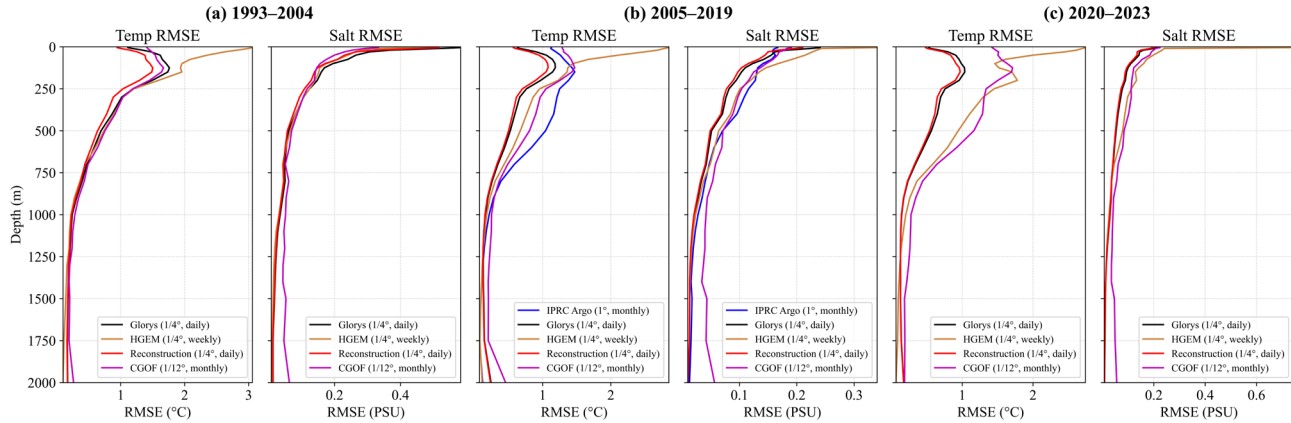

**Figure 15: RMSE profiles of different temperature and salinity datasets compared with WOD T-S profiles during (a) 1993–2004, (b) 2005–2019, and (c) 2020–2023.**

As shown in Fig. 15, the variation of RMSE with depth indicates that, throughout the entire evaluation period, the reconstructed temperature consistently exhibits the lowest RMSE at all depth levels compared with other temperature datasets, demonstrating high reliability.

For salinity, during 1993–2004, the CGOF salinity data show a slight advantage at depths shallower than 130 m; however, below 130 m, the reconstructed salinity data remain more consistent with the WOD salinity profiles (Fig. 15a). During

2005–2019, the CGOF dataset achieves the lowest RMSE near the surface (50 m), while at all other depths, the reconstructed salinity data better match the in situ observations (Fig. 15b).

Overall, the reconstructed temperature and salinity fields exhibit high reliability and stability across all depth layers, further confirming the robustness and accuracy of the proposed reconstruction approach.


To more intuitively illustrate the relationships between various temperature–salinity datasets and the WOD temperature–salinity profiles, four representative depths (50 m, 200 m, 800 m, and 1000 m) were selected. At each depth, density scatter plots were drawn for the reconstructed temperature–salinity data, as well as for the GLORYS2V4, HGEM, IPRC Argo, and CGOF datasets, in comparison with the WOD in situ observations. The density scatter plots of temperature data are

presented in Fig. 16, while those of salinity data are shown in Fig. 17.



**Figure 16: Density scatter plots between different temperature datasets and WOD observed temperature profiles at four representative depths (50 m, 200 m, 800 m, and 1000 m) over the period of 1993-2023.**



**Figure 17: Density scatter plots between different salinity datasets and WOD observed salinity profiles at four representative depths (50 m, 200 m, 800 m, and 1000 m) over the period of 1993-2023.**

From the density scatter distributions shown in Fig.16 and Fig. 17, the correlations and systematic bias characteristics between each dataset and the WOD in situ observations at different depths can be clearly identified. The reconstructed results proposed in this study exhibit the highest consistency and the most compact distribution across all four representative depths (50 m, 200 m, 800 m, and 1000 m). The scatter clouds are densely concentrated along the 1:1 diagonal


line, with distinct density peaks and narrow bias distributions, indicating that the model can accurately reproduce the real oceanic thermohaline structure on a large scale.

At the surface layer (50 m), influenced by air–sea exchanges and seasonal mixing, all datasets show a certain degree of dispersion. However, the reconstructed results demonstrate the smallest dispersion range, with almost no systematic 435 overestimation or underestimation. In contrast, GLORYS2V4 and CGOF exhibit slight offsets in regions of high temperature and salinity, reflecting weaker responses in strongly variable areas. At 200 m depth, the relationship between the reconstructed results and the WOD observations becomes more linear, suggesting that the model effectively captures the thermohaline gradient structure near the upper boundary of the thermocline. The HGEM and Argo products show slightly more scattered distributions at this depth, indicating moderate biases in reproducing the upper-to-mid-layer 440 thermohaline structure.

In the intermediate and deep layers (800 m and 1000 m), the reconstructed results maintain excellent agreement with the observations, with highly concentrated scatter and very low RMSE values. This demonstrates that the model performs not only well at the surface but also robustly reconstructs deep thermohaline structures. In comparison, GLORYS2V4 and 445 CGOF exhibit slightly broader scatter distributions, showing a systematic underestimation trend, while HGEM displays increased dispersion, reflecting weaker deep-layer constraints.

Overall, the density scatter distributions for both temperature and salinity confirm that the reconstructed data have the highest correlation and lowest errors relative to the WOD observations, maintaining consistent performance across all 450 depths. These results verify the high accuracy and robustness of the proposed reconstruction method under multi-scale and multi-depth conditions. Furthermore, they indicate that the developed neural network model effectively integrates multi-source information during the long-term reconstruction process—capturing rapid upper-ocean variability while accurately restoring deep-layer equilibrium structures—thus providing a solid foundation for high-quality three-dimensional thermohaline reanalysis.

**4 Data availability**

The data used in this study are available for consultation as described in Section 2.1.

The temperature–salinity data reconstructed using the method proposed in this study are freely accessible at https://doi.org/10.57760/sciencedb.31950 (Wang et al., 2025).



## 5 Discussion and Conclusion

To achieve real-time and accurate reconstruction of daily three-dimensional temperature and salinity fields (26 layers, 1/4°, 5–2000 m) in the northwestern Pacific region (0–40°N, 120–160°E), this study proposes an attention-enhanced 3D U-Net++ reconstruction framework that relies solely on real-time available SST and SSH data. The attention enhanced 3D U-Net++ architecture provides a mechanism for cross-scale feature aggregation and selective information gating through deep skip connections. The attention gates highlight temporal surface features that are more strongly correlated with the

target depth layers along the multi-resolution pathways, thereby mitigating noise propagation and over-smoothing. Combined with the 3D encoder–decoder structure, this design enhances the preservation of vertical gradients and frontal structures in the reconstructed fields. The network takes 26 consecutive days of SST and SSH as input, which effectively alleviates the intrinsic challenge of mapping from limited surface inputs to full-depth outputs. A single-time surface-to-volume mapping is inherently underdetermined, resembling a super-resolution problem. By incorporating inter-day

evolution, the model can extract implicit constraints associated with subseasonal variability, circulation, and mixing processes from the temporal continuity of surface signals, thereby reducing non-uniqueness and improving the discriminability of vertical structures. In addition, a transfer learning strategy is employed. The model is first pretrained using monthly SST/SSH and IPRC-Argo datasets to learn observation-dominated low-frequency spatiotemporal signals and stable structures. It is then retrained using daily SST/SSH and GLORYS2V4 data to capture the dynamic processes

and assimilation mechanisms embedded in the reanalysis fields. This two-stage approach establishes a balance between observational consistency and physical–dynamical constraints.

Analysis results demonstrate that, both on the validation set and in long-term statistical evaluations, the reconstructed temperature and salinity fields are more consistent with in situ observation profiles, verifying the reliability and accuracy

of the proposed method for subsurface ocean data inversion. The data obtained in this study not only provide important support for revealing mesoscale ocean dynamic processes but also lay a solid foundation for quantitatively estimating key elements such as ocean mass transport, heat, and salinity fluxes. In the future, these data hold broad application prospects and scientific value in advancing the study of mesoscale ocean dynamics, improving ocean model simulations, and enhancing the accuracy of climate predictions.

**Author contributions.**

HW and LZ conceptualized the study and contributed to the writing and revision of the manuscript. HW performed data processing, coding, and underwater temperature–salinity reconstruction. LZ, SY, and XY contributed to the discussion of results and critically reviewed the manuscript. ZL was responsible for data collection and partial code development.



**Competing interests.**

The contact author has declared that none of the authors has any competing interests.

**Acknowledgement.**

The authors gratefully acknowledge the use of the Optimum Interpolation Sea Surface Temperature (OISST) data provided by NOAA (https://www.ncei.noaa.gov/data/sea-surface-temperature-optimum-interpolation/v2.1/access/avhrr/), and the sea level data from the Archiving, Validation, and Interpretation of Satellite Oceanographic data (AVISO), available via

Copernicus Europe's eyes on Earth (https://data.marine.copernicus.eu/product/SEALEVEL_GLO_PHY_L4_MY_008_047/description). We also acknowledge the GLORYS2V4 reanalysis data for temperature and salinity gridded fields (https://data.marine.copernicus.eu/product/GLOBAL_MULTIYEAR_PHY_ENS_001_031/description) and the Argo monthly temperature and salinity gridded datasets provided by the International Pacific Research Center (IPRC,

http://apdrc.soest.hawaii.edu/projects/Argo/data/gridded/On_standard_levels/index-1.html).

Additional thanks are extended to the World Ocean Database for temperature–salinity profiles (https://www.ncei.noaa.gov/products/world-ocean-database), the Oceanographic Data Center, Chinese Academy of Sciences (CODC, https://www.casodc.com/), for the High-Resolution Northwest Pacific Temperature Salinity Current Dataset, and the National Oceanic Cloud (https://www.cmoc-china.cn) for providing the China Global Ocean Fusion

Dataset 1.0.

This work was supported by the Oceanographic Data Center, Institute of Oceanology, Chinese Academy of Sciences (IOCAS).

**Financial support.**

This study was supported by the National Natural Science Foundation of China (42576027), the Project of Science and

Technology Innovation of Laoshan Laboratory (LSKJ202201702), and the TS Scholar Program (tsqn202103128).

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
