# Peer review of "Attention Enhanced 3D-U-Net++ Ocean Temperature and Salinity Reconstruction in the Northwestern Pacific based on Transfer Learning"

_Earth System Science Data, 2025_

## Referee Comment (RC1)

This paper presents a 3D reconstruction method of ocean temperature and salinity based on Attention Enhanced 3D-U-Net++ and Transfer Learning for the Northwest Pacific region, using real-time Sea Surface Temperature (SST) and Sea Surface Height (SSH) data to generate a daily high-resolution (1/4°, 5-2,000 m depth) temperature-salinity field. The methods, despite incorporating designs such as attention and transfer learning, are those that have been widely used by previous authors and lack substantial innovation. Overall, this study requires significant refinement in terms of methodological detail, generalizability, and interpretability. I recommend a Major Revision.

1) In line 60 of the article, it is mentioned that the data assimilation has the problem of "there remain significant challenges in accurately reproducing the vertical structures of mesoscale eddies", but in this study, the reanalysis products based on the numerical model and data assimilation are used as the labels for training in fine-tuning stage, so is it possible that the problem of "inaccuracy of the vertical structure of mesoscale eddies" also exists in the present dataset? Please add extensive experimental analyses to explain how this study used "inaccurate" reanalysis products as labels to train the model to obtain "accurate" 3D thermohaline fields?

2) Table 1 appears to have a non-English "、"。

3) The combination of UNet and CBAM does not have novelty; many studies have been carried out by previous researchers [1], [2], [3], and this paper does not have a substantial improvement and is not innovative enough.

4) The two-stage transfer learning is equally uninspiring. Combined with Fig. 4, the reconstruction accuracy is improved by less than 10% after transfer learning, and the result is not listed in the table with detailed values; is it intentionally avoided? Meanwhile, as shown in Figure 9, the reconstruction results are almost no different from GLORYS, especially the salinity reconstruction results, and the improvement of reconstruction accuracy is extremely limited. Based on this, is it necessary to carry out the process of such a complex reconstruction? Is it possible

to achieve better results with more detailed model tuning? Or is it possible to train to a higher accuracy by replacing the training labels with reanalysis products that have a higher accuracy than GLORYS2V4?

5) Why is it straightforward to say that inputting 26 days is optimal without any ablation experiments for other time periods, such as 2, 4, 6, 15, etc., up to 100 days, and is it not necessary to take into account the temporal correlation of the thermohaline high elements? Please analyze the temporal correlation of temperature and salt elements in this sea area with historical data, and also add ablation experiments for multiple days, and analyze the results of the experiments against the temporal correlation, so as to make a strong case that 26 days is the optimal option.

6) What is the quality control method for the profiles described in line 330? The number of profiles selected, 7833, is much less than the number of original profiles. Was it an intentional effort to select profiles that favored this study? Also, in terms of spatial distribution, the profiles are not uniform, so large blank areas of the profiles are not assessable for reconstruction accuracy, so the dataset is not entirely credible. How can we verify the reconstruction accuracy of the model in regions with no or sparse profiles?

7) This paper has repeatedly emphasized that this dataset has the advantage of "real-time", so please add detailed information on the update cycles of various products, the hardware environment for model training and inference, and the time spent. At the same time, please list the update cycles of several mainstream ocean reanalysis products and real-time objective analysis products. By comparison, please illustrate the advantages of this dataset in "real-time".

8) This study lacks comparisons with other mainstream marine reanalysis products, such as HYCOM, ECCO2, ORA5, CORA2, SODA3, and so on.

9) Some sentences are too long and could be split to improve readability.

10) As shown in Figures 7 and 8, the reconstruction results are almost identical to the GLORYS reanalysis. However, the model inputs are sea surface temperature and height information, and not even sea surface salinity information. How to restore

so many small- and medium-scale details of the approximate reanalysis products with so little information? Please give a more detailed description of the training process.

References

[1] H. Xie, Q. Xu, Y. Cheng, X. Yin, and K. Fan, "Reconstructing three-dimensional salinity field of the South China Sea from satellite observations," Front. Mar. Sci., vol. 10, p. 1168486, /05/08 2023, doi: 10.3389/fmars.2023.1168486.

[2] H. Xie, Q. Xu, Y. Cheng, X. Yin, and Y. Jia, "Reconstruction of Subsurface Temperature Field in the South China Sea From Satellite Observations Based on an Attention U-Net Model," IEEE Trans. Geosci. Remote Sens., vol. 60, pp. 1–19, 2022, doi: 10.1109/TGRS.2022.3200545.

[3] J. Qi, B. Xie, D. Li, J. Chi, B. Yin, and G. Sun, "Estimating thermohaline structures in the tropical Indian Ocean from surface parameters using an improved CNN model," Front. Mar. Sci., vol. 10, Apr. 2023, doi: 10.3389/fmars.2023.1181182.